Head and neck squamous cell carcinoma-specific prognostic signature and drug sensitive subtypes based on programmed cell death-related genes

Xu Chengbo 1
Xu Hongfang 1
Liu Baimei dannis1995@163.com 2
1 Department of Otolaryngology Head and Neck Surgery, Jinhua Wenrong Hospital , Jinhua , China
2 Department of Otolaryngology Head and Neck Surgery, Yongkang First People’s Hospital , Yongkang , China
Xu Zhijie
Electronic publication date: 2023 Nov 21
Publication date: 2023
Volume: 11
Electronic Location ID: e16364
Received 2023 May 8; Accepted 2023 Oct 6
Copyright: ©2023 Xu et al.
Copyright year: 2023
Copyright holder: Xu et al.
License: This is an open access article distributed under the terms of the Creative Commons Attribution License, which permits unrestricted use, distribution, reproduction and adaptation in any medium and for any purpose provided that it is properly attributed. For attribution, the original author(s), title, publication source (PeerJ) and either DOI or URL of the article must be cited.
License URL: https://creativecommons.org/licenses/by/4.0/

Keywords: Programmed cell death, Classification, Prognostic signature, Immunotherapy, Chemotherapy, Head and neck squamous

Funding: The authors received no funding for this work.

==============================
Background

As a complex group of malignancies, head and neck squamous cell carcinoma (HNSC) is one of the leading causes of cancer mortality. This study aims to establish a reliable clinical classification and gene signature for HNSC prognostic prediction and precision treatments.

Methods

A consensus clustering analysis was performed to group HNSC patients in The Cancer Genome Atlas (TCGA) database based on genes linked to programmed cell death (PCD). Differentially expressed genes (DEGs) between subtypes were identified using the “limma” R package. The TCGA prognostic signature and PCD-related prognostic genes were found using a least absolute shrinkage and selection operator (LASSO) regression analysis and univariate Cox regression analysis. The robustness of the LASSO analysis was validated using datasets GSE65858 and GSE41613. A cell counting kit-8 (CCK-8) test, Western blot, and real-time reverse transcriptase-polymerase chain reaction (RT-qPCR) were used to evaluate the expression and viability of prognostic genes.

Results

Four molecular subtypes were identified in PCD-related genes. Subtype C4 had the best prognosis and the highest immune score, while subtype C1 exhibited the most unfavorable outcomes. Three hundred shared DEGs were identified among the four subtypes, and four prognostic genes (CTLA4, CAMK2N1, PLAU and CALML5) were used to construct a TCGA-HNSC prognostic model. High-risk patients manifested poorer prognosis, more inflammatory pathway enrichment, and lower immune cell infiltration. High-risk patients were more prone to immune escape and were more likely to be resistant to Cisplatin and 5-Fluorouracil. Prognosis prediction was validated in external datasets. The expression of CTLA4, CAMK2N1, PLAU and CALML5 was enhanced in CAL-27 and SCC-25 cell lines, and CALML5 inhibited CAL-27 and SCC-25 cell viability.

Conclusion

This study shares novel insights into HNSC classification and provides a reliable PCD-related prognostic signature for prognosis prediction and treatment for patients with HNSC.

Introduction

A previous study analyzed the incidence rate of head and neck squamous cell carcinoma (HNSC) at the lip, oral cavity (377,713 new cases/2.0% of all cases), larynx (184,615/1.0%), salivary glands (53,583/0.3%), oropharynx (98,412/0.5%), nasopharynx (133,354/0.7%) and hypopharynx (84,254/0.4%) in 185 countries (Sung et al., 2021). Patients with HNSC are frequently diagnosed at an advanced stage with unfavorable clinical outcomes and surgery can only be performed in patients with resectable tumors, according to the tumor-node-metastasis (TNM) staging system. Unfortunately, it is usually difficult to identify resectable tumors in locally advanced HNSC, and only patients with advanced HNSC at the oropharynx and larynx benefit from surgery, with surgery not considered beneficial for patients with other types of advanced HNSC (Park et al., 2021). The five-year survival rate for patients with locally advanced HNSC is approximately 50% after standard treatment of concurrent chemoradiotherapy (Powell et al., 2020). It is therefore important to identify potential biomarkers to help predict clinical outcomes and provide precision therapeutics for HNSC patients.

Programmed cell death (PCD), which is a kind of cell death regulated by multiple biomacromolecules, is different from accidental and uncontrolled cell death (Peng et al., 2022). Some well-known types of PCD include alkaliptosis necroptosis, cuproptosis, parthanatos, pyroptosis, apoptosis, oxeiptosis NETosis, autophagy-dependent cell death, ferroptosis, entosis, and lysosome-dependent cell death (Bedoui, Herold & Strasser, 2020; Wang & Kanneganti, 2021). Studies on PCD have been widely conducted due its significant role in many severe diseases. Evidence supports the potential role of several signaling pathways of PCD in regulating the tumor microenvironment (TME) in cancer (Liu et al., 2022) and in predicting clinical outcomes (Tong et al., 2022). One recent study demonstrated that cuproptosis-associated lncRNAs can be used as predictive biomarkers for patients with HNSC (Yang et al., 2022). A ferroptosis-related DNA methylation signature can also serve as a risk classifier for patients with HNSC (Xu et al., 2022). Integrating ferroptosis and pyroptosis-related genes may provide a novel method for indicating prognosis and TME in HNSC patients (Yu et al., 2022). Inducing apoptosis in cancer cells is the goal of many cancer treatments, but the presence of oncogenic alterations in these cells and a disturbed tumor microenvironment both significantly limit apoptosis induction. Targeted and individualized treatments are therefore needed that consider the specific advantages and disadvantages of individual tumors (Raudenská, Balvan & Masařík, 2021). A comprehensive analysis of PCD-associated genes in predicting prognosis and therapeutic response in HNSC would assist in the advancement of those targeted treatments.

This study comprehensively identified four HNSC subtypes based on PCD-related genes and constructed a PCD-related prognostic signature using four PCD-related prognostic genes (CTLA4, CAMK2N1, PLAU and CALML5) in The Cancer Genome Atlas (TCGA)-HNSC database. Model robustness was then validated in the GSE65858 and GSE41613 datasets. This study provides novel insights into the underlying mechanisms of PCD-related genes in HNSC and in risk classification and therapeutic options for guiding immunotherapy and chemotherapy in patients with HNSC.

Materials and Methods

Data collection and processing

The TCGA GDC API was used to retrieve RNA-Seq data from HNSC samples in the TCGA database (https://portal.gdc.cancer.gov/) using SangerBox bioinformatics tools (http://sangerbox.com/ (Shen et al., 2022)). This study included a total of 499 primary HNSC tumor samples and 44 para-carcinoma tissues. Samples that lacked clinical follow-up information or status were excluded from further examination. Samples with a survival time of longer than 30 days were included in the study. Ensembl gene IDs were converted to gene symbol IDs, and the average expression of genes with numerous gene symbol IDs was calculated. The expression data of GSE65858 (Wichmann et al., 2015), which included 253 HNSC samples, and GSE41613 (Zhao et al., 2022), which included 97 HNSC samples, were taken from the Gene Expression Omnibus (GEO; https://www.ncbi.nlm.nih.gov/geo/) database (Barrett et al., 2013). To pre-process data from the GEO database, annotation information was retrieved to map probes to genes using annotation files. When numerous probes matched the same gene, the expression of that gene was expressed as an average value. The TCGA-HNSC dataset was used as a training set in this study, and the GSE65858 and GSE41613 datasets were used as independent validation datasets. Twelve PCD patterns (apoptosis, necroptosis, pyroptosis, ferroptosis, cuproptosis, entotic cell death, netotic cell death, parthanatos, lysosome-dependent cell death, autophagy-dependent cell death, alkaliptosis, and oxeiptosis) were obtained from a previous study (Zou et al., 2022).

Relationship between PCD and clinicopathologic features

The 12 PCD patterns from the TCGA-HNSC dataset were scored using a single-sample gene set enrichment analysis (ssGSEA) in the gene set variation analysis (GSVA) package (version 1.34.0, https://bioconductor.riken.jp/packages/3.0/bioc/html/GSVA.html) (Hänzelmann, Castelo & Guinney, 2013) and their differences between tumorous and para-carcinoma tissues were analyzed. The expressions of PCD-related genes between tumor and para-carcinoma tissues were also evaluated to screen differentially expressed PCD genes. Then, the distribution of PCD gene signatures in different clinicopathologic feature groups (T stage, N stage, M stage, stage, grade, and gender) was assessed.

Identification of molecular subtypes based on PCD-related genes

The association of PCD-related gene expression with HNSC prognosis was analyzed using a univariate Cox analysis based on the “survival” R package (https://bioconductor.org/packages/devel/bioc/vignettes/survtype/inst/doc/survtype.html), and genes significantly associated with prognosis were identified using a P value <0.001. The “ConsensusClusterPlus” R package (version 3.17, https://bioconductor.org/packages/release/bioc/html/ConsensusClusterPlus.html; Wilkerson & Hayes, 2010) was used to perform a consensus clustering analysis on 26 PCD-related genes significantly related to prognosis. A total of 500 bootstraps containing 80% TCGA-HNSC samples were processed with a km algorithm and 1-Spearman correlation. The number of clusters was 2–10. The consensus matrix and consensus cumulative distribution function (CDF) were used to find the optimal subtypes. The TCGA dataset was used to construct Kaplan–Meier curves of overall survival (OS) and progression-free survival (PFS). Clinicopathologic variables (grade, gender, age, T stage, N stage, stage) were included and analyzed for variations in clinicopathologic features among TCGA-HNSC molecular subtypes.

Genomic heterogeneity among subtypes

Molecular characteristics of TCGA-HNSC were obtained from a previous pan-cancer study (Thorsson et al., 2018) to evaluate genomic heterogeneity among subtypes. Previously identified subtypes of HNSC (atypical, mesenchymal, basal, and classical), as outlined by The Cancer Genome Atlas Network (2015) were compared with the molecular subtypes identified in this study.

The DEGs among subtypes

A differential expression analysis was performed using the “limma” R package (version 3.42.2, https://bioconductor.org/packages/release/bioc/html/limma.html; (Ritchie et al., 2015)) comparing each subtype with all other subtypes. DEGs were defined if the false discovery rate (FDR) <0.05 and —log2FC—>log2(1.5). Through overlapping in a Venn diagram, the DEGs were filtered at the intersection of C1 vs other subtypes, C2 vs other subtypes, C3 vs other subtypes, and C4 vs other subtypes.

Functional enrichment analysis

A GSEA for molecular subtypes was performed using the Kyoto Encyclopedia of Genes and Genomes (KEGG) dataset from the MSigDB database (https://ngdc.cncb.ac.cn/databasecommons/database/id/1077; Liberzon et al., 2015). Among these subtypes, pathways with FDR<0.05 were considered differentially activated. To assess the biological processes involved, the “clusterProfiler” R package (version 3.14.3, https://bioconductor.org/packages/release/bioc/html/clusterProfiler.html; Yu et al., 2012) was used to perform a functional enrichment analysis. The Database for Annotation Database for Annotation, Visualization and Integrated Discovery (DAVID, https://david.ncifcrf.gov/; Dennis Jr et al., 2003) was used to perform a KEGG pathway enrichment analysis for the DEGs.

Protein–protein interaction (PPI) network analysis

A PPI network was constructed in the STRING database (https://cn.string-db.org/) (Szklarczyk et al., 2019) with confidence scores ≥ 0.7. The constructed network was then downloaded and then visualized using the Cytoscape software (version 3.7.2; Shannon et al., 2003).

Development and verification of the prognostic signature

A univariate Cox regression analysis based on the DEGs was employed to select genes showing significant influence on HNSC prognosis under the threshold of P <0.05 in the ‘survival’ R package (https://bioconductor.org/packages/devel/bioc/vignettes/survtype/inst/doc/survtype.html). To prevent overfitting issues, the “glmnet” R package (version 4.1-2, https://cran.r-project.org/web/packages/glmnet/index.html) was used in the LASSO analysis (Hastie, Qian & Tay, 2021). Important PCD-related genes were then optimized using a stepwise multivariate regression analysis with stepAIC. The following formula was used to calculate the risk score: Risk score = ∑βi*ExPi, where i and β represent expression level and coefficient value from the Cox regression analysis, respectively.

Each sample in the TCGA-HNSC training set was assigned a z-scored risk score. HNSC samples were split into two categories, high and low risk, based on a zero-risk threshold. Overall survival of high- and low-risk patients was compared using Kaplan–Meier curves; the “survival” R package (version 3.2-10, https://cran.r-project.org/web/packages/survivalAnalysis/index.html) was used for statistical analysis; the “survminer” R package (version 0.4.9, https://cran.r-project.org/web/packages/survminer/index.html) was used for visualization; and a receiver operating characteristic (ROC) analysis was performed using the “timeROC” R package (version 0.4, https://cran.r-project.org/web/packages/timeROC/index.html; Blanche, Dartigues & Jacqmin-Gadda, 2013). The robustness of the risk model was further tested in validation datasets.

Analysis of tumor microenvironment (TME)

In TCGA-HNSC, the TME difference of HNSC patients with various molecular subtypes or risk was examined according to immune cell infiltration reflected by the level of genes in immune cells. The ESTIMATE package (version 1.0.13, https://bioinformatics.mdanderson.org/estimate/index.html) was used to analyze immune cell infiltration. CIBERSORT (http://CIBERSORT.stanford.edu/; Craven, Gökmen-Polar & Badve, 2021) was used to calculate the percentages of 22 immune cells based on gene expression profiles (Chen et al., 2018). MCP-counter (Becht et al., 2016), TIMER (http://timer.comp-genomics.org/timer/; Li et al., 2017), and EPIC (Racle & Gfeller, 2020) in the IOBR R package (version 0.99.9, https://github.com/IOBR/IOBR) were further employed to evaluate multiple immune cell infiltrations. Differences in inflammatory pathways, such as the NF-kappa B signaling pathway, the JAK-STAT signaling pathway, the toll-like receptor signaling pathway, the cGAS-STING signaling pathway, and the MAPK signaling pathway, were also evaluated. The correlation between TME and the expression of risk genes in the model was analyzed. TISCM2 (http://tisch1.comp-genomics.org/) was used to analyze the expression patterns of CTLA4, CAMK2N1, PLAU, and CALML5 in HNSC cells.

Prediction of response to immunotherapy

A total of 79 immunological checkpoints were obtained using the methods outlined by Hu et al. (Maeser, Gruener & Huang, 2021), and their expression patterns were evaluated and visualized using a heatmap to predict their response to immunotherapy. The outcome of immune checkpoint inhibition (ICI) treatment was also predicted using the TIDE algorithm (http://tide.dfci.harvard.edu/; Tschöpe et al., 2021). A high TIDE score predicted a greater risk of immunological escape and less therapeutic effectiveness.

Drug sensitivity analysis

The sensitivity of HNSC patients to small molecular therapeutic drugs was evaluated based on the IC50 value for each drug using the “oncoPredict” R package (version 2.0, https://cran.r-project.org/web/packages/oncoPredict/index.html; Maeser, Gruener & Huang, 2021). A correlation analysis between risk score and estimated IC50 value was performed and drugs with FDR <0.05 and —correlation—>0.4 were considered to be significantly related to risk score.

Cell culture and transient transfection

The Beina Biotechnology Institute (Beijing, China) donated the human oral squamous cell carcinoma (OSCC) cell lines CAL-27 and SCC-25 and the human oral epithelial cells (HOEC) used in this study. The cells were grown in F12 DMEM media containing 10% fetal bovine serum. Both cell lines were kept in a humid incubator at 37 °C with 5% CO2.

Lipofectamine 2000 (Invitrogen, Carlsbad, CA, USA) was used to transfect CALML5 siRNA (Sigon, China) or ovexpression-CALML5 (oe-CALML5, Sigon, China) into the cells. The CALML5 siRNA target sequence was CGCATTGATCTAAATAAAGGACT (CALML5-si). The target sequences for oe-CALML5 were F: ATACTCGAGCGGGTTGACACGGATGGAAACG (5′–3′) and R: ATAGC GGCCGCACTCCTGGAAGCTGATTTCGC (5′–3′; oe-CALML5).

RT-qPCR

TRIzol reagent (Sigma-Aldrich, St. Louis, MO, USA) was used to isolate total RNA from the CAL-27, SCC-25, and HOEC cell lines. M-MLV reverse transcriptase and RiboLock nucleic acid enzyme inhibitor (Applied Biosystems, Waltham, MA, USA) were employed to reverse-transcribe RNA into cDNA. RT-qPCR was performed using 2 ug RNA from each sample, a Roche ABI Q5 PCR System (Applied Biosystems, Waltham, MA, USA), and FastStart SYBR Green Master (TaKaRa, Dalian, China). Total reaction volume was 20 ul, which consisted of cDNA coupled with 0.5 ul of forward and reverse primers, 2 ul of cDNA template, and enough water to reach a total volume of 20 ul. The 2−ΔΔC(T) method was used to analyze relative changes in gene expression from the RT-qPCR experiments (Livak & Schmittgen, 2001). The sequences of primer pairs for the target genes are shown in Table 1.

Western blot

The proteins of the CAL-27, SCC-25, and HOEC cell lines were lysed by RIPA buffer (Solarbio, Beijing, China) and denatured at 100 °C for 15 min. Protein extracts from the samples were run through 10% SDS-PAGE, and the separated proteins were then transferred to PVDF membranes. After blocking the membranes with a 5% solution of skim milk powder for an hour, the membranes were incubated overnight with primary antibodies, such as anti-CALML5 antibody (1:1000, SAB1401711; Sigma-Aldrich, Waltham, MA, USA) and anti-GADPH antibody (1:1000, Proteintech, 60004-1-Ig). The membranes were then exposed to secondary antibodies (Anti-Rabbit IgG: 1:4000; ab150077; Abcam, Cambridge, MA, USA) for an hour. Protein signals were captured by a hypersensitive ECL Chemiluminescence Detection Kit (Proteintech Group, Chicago, IL, USA) and analyzed using densitometric analysis software Quantity One v4.6.2 for Windows (Bio-Rad, Hercules, CA, USA).

Cell viability

CCK-8 (Beyotime, Beijing, China) was performed to determine cell viability. In 96-well plates, treated cells were cultured at a density of 1 ×105 cells per well. The CCK-8 solution was used at 12 h, 24 h, 36 h and 48 h. The OD450 of each well were measured using a microplate reader (Tecan Infinite M200 Micro Plate Reader; LabX, Männedorf, Switzerland) after two hours of incubation at 37 °C.

Table 1 Gene primers.

Gene	Forward primer sequence (5–3)	Reverse primer sequence (5–3)	
CTLA4	GGCTTGCCTTGGATTTCAGC	GAGGGCTTCTTTTCTTTAGCAATTA	
CAMK2N1	CGGAGCAAGCGGGTTGTTAT	CACAACAGACTGCAAGGGGA	
PLAU	CCAAAGTGAGTGCGCTCTTG	CAGTCACAGTTCGCCTGTTC	
CALML5	GGTTGACACGGATGGAAACG	ACTCCTGGAAGCTGATTTCGC	
GAPDH	AATGGGCAGCCGTTAGGAAA	GCCCAATACGACCAAATCAGAG	

Statistical analyses

R software was used for data processing and analysis (version 3.6.3). A Venn diagram was plotted in VennDiagram (version 1.6.20, https://cran.r-project.org/web/packages/VennDiagram/index.html), and a principal component analysis was performed using ggbiplot (version 0.55, https://bioconductor.org/help/search/index.html?q=ggbiplot/). FDR was corrected using the Benjamin and Hochberg method (Smith et al., 2016). Significant differences in survival analyses were defined using Kaplan–Meier and log-rank tests. Two-group difference was assessed by a Wilcox test, while the Kruskal-Wallis test followed by Dunn’s post hoc test were used to assess differences in more than two groups. The Chi-square test was used to examine the distribution of clinicopathologic characteristics in different subtypes or risk groups. The area under the ROC curve (AUC) was used to test predictive performance (AUC of 0.9–1.0: excellent; 0.8–0.9: very good; 0.7–0.8: good; 0.6–0.7: sufficient, 0.5–0.6: bad; and <0.5: not useful; (Fluss, Faraggi & Reiser, 2005; Zhang et al., 2022c). A correlation analysis was performed with the Spearman test. A p < 0.05 was regarded as a statistically significant difference.

Results

Relationship between PCD and clinicopathologic features of HNSC patients

The study workflow is shown in Fig. S1. The differences of 12 PCD gene signatures were compared between tumorous and para-carcinoma tissues. The results showed that most of the PCD gene signatures were enriched in HNSC tumor samples (Fig. 1A). The expression level of PCD-related genes in tumor and para-carcinoma tissues was calculated (Fig. 1B). A total of 166 differentially expressed PCD-related genes were obtained, and their biological processes were revealed using a functional enrichment analysis. Fig. 1C displays significant enrichment of these genes in cell death-related pathways. Several PCD gene signatures were noticeably altered between early- and late-stage HNSC as well as between low- and high-grade tumors. Parthanatos, pyroptosis, apoptosis, necroptosis, autophagy, cuproptosis, ferroptosis, lysosome dependent cell death, and entotic cell death were increased in female patients (Fig. 1D).

Figure 1 Relationship between PCD and clinicopathologic features of TCGA-HNSC patients.

(A) Differences of PCD gene signatures between tumor and para-carcinoma tissues. (B) Differential expression analysis of PCD-related genes between tumor and para-carcinoma tissues. (C) KEGG enrichment analysis for differential expressed PCD-related genes. (D) Box plots of PCD gene signatures between different TMN stage, stage, grade and gender groups. Ns represents P > 0.05. ∗P < 0.05; ∗∗P < 0.01; ∗∗∗P < 0.001; ∗∗∗∗ P < 0.0001.

Four molecular subtypes identified based on PCD-correlated genes

The association of PCD-related gene expression with HNSC prognosis was analyzed and a total of 26 PCD-related genes that demonstrated great influence on prognosis were discovered, including 19 apoptosis-related genes and nine pyroptosis-related genes (Fig. S2). A consensus clustering analysis was conducted to determine molecular subtypes. CDF curves were relatively stable when cluster = 4 (Figs. S3A–S3B). Four molecular subtypes were classified when consensus matrix k = 4 (Fig. S3C). Figure 2A shows that these four subtypes were distinctly separated. Kaplan–Meier curves were generated for a survival analysis. Subtype C4 had the best prognosis, while subtype C1 exhibited the poorest prognosis (Figs. 2B–2C). Most of the PCD gene signatures, except for parthanatos, were significantly different among subtypes (Figs. 2D–2E). Moreover, subtype C1 exhibited higher grade tumors (Fig. 2F).

Figure 2 Four molecular subtypes are identified based on PCD-related genes TCGA-HNSC dataset.

(A) Principal components analysis (PCA) shows distinct separation among subtypes. (B) Kaplan–Meier curves of OS for four subtypes. (C) Kaplan–Meier curves of PFS for four subtypes. (D–E), The expression patterns of PCD-related genes among different subtypes. (F) Distribution of clinicopathologic features in different subtypes. Ns represents P > 0.05. ∗P < 0.05; ∗∗∗∗ P < 0.0001.

Genomic heterogeneity among subtypes

The genomic heterogeneity among subtypes was investigated in TCGA-HNSC. As shown in Figs. 3A–3I, subtype C1 possessed higher homologous recombination defects, aneuploidy score, number of segments, and fraction altered, whereas subtype C2 had increased nonsilent mutation rate and single-nucleotide variant (SNV) neoantigens than other subtypes. Most patients in subtype C1 belonged to the basal HNSC subtype, while more than half of the patients in subtype C4 had atypical HNSC (Fig. 3J).

Figure 3 Genomic heterogeneity among subtypes in TCGA-HNSC dataset.

(A–I) Violin plots of molecular characteristics among four subtypes. (J), Association of identified four subtypes and previously reported subtypes (atypical, mesenchymal, basal and classical). Ns represents P > 0.05. ∗P < 0.05; ∗∗P < 0.01; ∗∗∗P < 0.001; ∗∗∗∗ P < 0.0001.

Changes in the TME among subtypes

To evaluate changes in the TME among subtypes, ESTIMATE was used to analyze immune cell infiltration (Fig. 4A). Subtype C4 had the highest ImmuneScore of the subtypes. As displayed in Fig. 4B, Macrophages_M1, T_cells_CD4_memory_activated, and T_cells_CD8 were all enriched in subtype C4. MCP-counter, TIMER, and EPIC results revealed that fibroblasts were significantly abundant in subtype C1 (Figs. 4C–4E). Differences in inflammatory pathways were compared, and the results showed increased inflammatory pathway scores in subtype C4 (Fig. 4F).

Figure 4 Changes of tumor microenvironment (TME) among four subtypes in TCGA-HNSC dataset.

(A) Box plots of immune scores estimated by ESTIMATE. (B) Box plots of 22 immune cells estimated by CIBERSORT. (C–E) Box plots of immune cells estimated by MCP-counter, TIMER and EPIC. (F) Violin plots of inflammatory pathways among four subtypes. Ns represents P > 0.05. ∗P < 0.05; ∗∗P < 0.01; ∗∗∗P < 0.001; ∗∗∗∗ P < 0.0001.

Relationship between molecular subtypes and immunotherapy

The expressions of immune checkpoints were subsequently analyzed among subtypes. Figure 5A shows that most of the immune checkpoints were highly-expressed in subtypes C3 and C4 but had low expression levels in subtypes C1 and C2. The expression levels of PDCD1 (PD-1), CTLA4, and CD274 (PD-L1) were detected in all subtypes (Fig. 5B), and the results found that all three were significantly elevated in subtype C4 compared with the other subtypes. TIDE was used to predict the response of subtypes to immunotherapy. Subtype C4 exhibited the highest TIDE score, indicating that escape from immunotherapy was more likely to occur in patients with subtype C4. Among the different molecular subtypes in TCGA-HNSC, the highest exclusion score, TIDE was in subtype C1, indicating C1 is more suitable for immunotherapy (Fig. 5C).

Figure 5 Prediction of response to immunotherapy among subtypes in TCGA-HNSC dataset.

(A) The expression patterns of immune checkpoints among subtypes. (B) Violin plots of the expression of PDCD1 (PD-1), CTLA4 and CD274 (PD-L1) among subtypes. (C) Violin plots of the TIDE, dysfunction and exclusion among subtypes. Ns represents P > 0.05. ∗P < 0.05; ∗∗ P < 0.01; ∗∗∗P < 0.001; ∗∗∗∗P < 0.0001.

Pathway characteristics analysis among subtypes

To determine whether there were differentially activated pathways in different molecular subtypes, hallmark genes were acquired for GSEA. Several proliferation-related pathways, including HALLMARK_G2M_CHECKPOINT, HALLMARK_MYC_TARGETS_V2, HALLMARK_P53_PATHWAY, HALLMARK_E2F_TARGETS, HALLMARK_MTORC1 _SIGNALING, and HALLMARK_MYC_TARGETS_V1 were significantly enriched in subtype C1 (Fig. 6A). Some immune-related pathways, such as HALLMARK _ALLOGRAFT_REJECTION, HALLMARK_INTERFERON_GAMMA_RESPONSE, HALLMARK_INTERFERON_ALPHA_RESPONSE, HALLMARK_COMPLEMENT, and HALLMARK_INFLAMMATORY_RESPONSE, were activated in subtypes C3 and C4.

Figure 6 Pathway characteristics analysis and identification of DEGs among subtypes in TCGA-HNSC dataset.

(A) Analysis of differential activated pathways using GSEA among subtypes. (B) Differential expression analyses for molecular subtypes. (C) Venn diagram shows the intersection among C1 vs other, C2 vs other, C3 vs other and C4 vs other. (D) Functional enrichment analysis of 300 DEGs.

Identification of DEGs among subtypes and their enriched pathways

To identify the DEGs among subtypes, the following differential expression analyses were performed: C1 vs. other subtypes, C2 vs. other subtypes, C3 vs. other subtypes, and C4 vs. other subtypes (Fig. 6B). There were 1,085 DEGs identified between subtype C1 and other subtypes (including 343 upregulated genes and 742 downregulated genes), 1,292 DEGs (1,080 downregulated genes and 212 upregulated genes) between subtype C2 and other subtypes, 852 DEGs (141 downregulated genes and 711 upregulated genes) between subtype C3 and other subtypes, and 1,404 DEGs (319 downregulated genes and 1,085 upregulated genes) between subtype C4 and other subtypes. The Venn diagram shown in Fig. 6C shows a total of 300 shared DEGs among the DEGs identified in the four differential expression analyses. These 300 DEGs were mainly enriched in immune-related pathways, comprising regulation of lymphocyte activation, Th17 cell differentiation, regulation of leukocyte activation, and T cell activation, as shown by functional enrichment analysis (Fig. 6D). A PPI analysis on the 300 DEGs detected several genes, including CTLA4 and PLAU, that interacted with each other (Fig. S4A). A functional enrichment analysis on the 300 DEGs using DAVID software showed that they were enriched in immune pathways and infection-related pathways (Fig. S4B).

Construction and validation of HNSC prognostic signature

A univariate Cox regression analysis of the 300 DEGs identified 126 genes that influenced HNSC prognosis, including 90 protective genes and 36 risk genes. A LASSO analysis further reduced the number of genes associated with HNSC prognosis. The trajectory of each independent variable with lambda is shown in Fig. 7A. Independent variable coefficients close to zero gradually increased with increased lambda, and then the confidence interval under each lambda was assessed using 3-fold cross validation (Fig. 7B). When lambda = 0.0528, six genes were identified. Stepwise multivariate regression analysis was used to determine genes for the HNSC prognostic signature, and four genes were finally selected (Fig. 7C).

Figure 7 Construction and validation of prognostic signature.

(A) The trajectory of each independent variable with lambda. (B) Confidence interval under each lambda. (C) Four genes are identified as PCD-related prognostic genes. (D) Distribution of RiskScore, survival time, survival status and PCD-related prognostic genes in TCGA-HNSC. (E), Kaplan–Meier curves with ROC curves in training dataset. (F–G) Distribution of RiskScore, survival time, survival status and PCD-related prognostic genes as well as Kaplan–Meier curves with ROC curves in GSE65858 dataset. (H–I) Distribution of RiskScore, survival time, survival status and PCD-related prognostic genes as well as Kaplan–Meier curves with ROC curves in GSE41613 dataset.

The risk core of each patient in the TCGA-HNSC training set was calculated using RiskScore =−0.234*CTLA4+0.14*CAMK2N1+0.095*PLAU−0.068*CALML5. HNSC patients considered high risk had a higher OS rate than that of low-risk patients, with a 1-year area under the curve (AUC) of 0.66, 3-year AUC of 0.7, and 5-year AUC of 0.62 (Figs. 7D–7E). The predictive value of the HNSC prognostic signature was then validated using the GSE65858 and GSE41613 datasets (Figs. 7F–7I), and a strong performance in HNSC prognosis prediction was observed.

Relationship between prognostic signature and clinicopathologic features

The association between the risk score and clinicopathological characteristics of TCGA-HNSC patients, including stage, T stage, N stage, grade, gender, age, and status, was investigated. Deceased patients showed higher risk scores (Figs. 8A–8G), and late-stage HNSC patients also had higher risk scores. Further analysis of the risk categories and molecular subtypes revealed that subtypes C1 and C2 accounted for the majority of high-risk patients and subtypes C3 and C4 accounted for the majority of low-risk patients. Subtype C1 had the highest risk score and C4 had the lowest risk score (Fig. 8H). Risk score also enhanced the expression of the four risk genes (CTLA4, CAMK2N1, PLAU, and CALML5; Fig. 8I).

Figure 8 Relationship between prognostic signature and clinicopathologic features in TCGA-HNSC dataset.

(A–G) Distribution of clinicopathological features (T stage, N stage, stage, grade, gender, age and status) between risk groups and violin plots of risk score between clinicopathological feature groups. (I) Association of four risk genes (CTLA4, CAMK2N1, PLAU and CALML5) with clinicopathological features. Ns represents P > 0.05. ∗∗∗∗P < 0.0001.

Immune characteristics of risk groups

Low-risk patients had higher ImmuneScores and ESTIMATE Scores than high-risk patients, indicating greater immune infiltration (Fig. 9A). The findings showed that CD8+T cells, M1 macrophages, and activated memory CD4+ T cells were enriched in low-risk patients, but resting memory CD4+ T cells CD4 and M0 macrophages were concentrated in high-risk patients (Fig. 9B). Fibroblasts were present in a large number of high-risk patients (Figs. 9C–9E). The toll-like receptor signaling pathway, the NF-kappa B signaling pathway, and the MAPK signaling pathway were all active in high-risk patients (Fig. 9F). Risk score and risk genes (CTLA4, CAMK2N1, PLAU, and CALML5) were both significantly correlated with considerable immune cells and inflammatory pathways (Fig. 9G). As shown in Fig. 9H, CALML5, CAMK2N1, CTLA4, and PLAU were highly-expressed in malignant HNSC cells, myofibroblasts cells, Treg cells, and Mono-Macro cells, respectively.

Figure 9 Immune characteristics of risk groups in TCGA-HNSC dataset.

(A) Box plots of immune scores estimated by ESTIMATE. (B) Box plots of 22 immune cells estimated by CIBERSORT. (C–E) Box plots of immune cells estimated by MCP-counter, TIMER and EPIC. (F) Violin plots of inflammatory pathways between risk groups. (G) Correlation analysis for risk score, risk genes and TME. (H) The expression patterns of CTLA4, CAMK2N1, PLAU and CALML5 in different HNSC cells. oral squamous cell carcinoma: OSCC, Nasopharyngeal carcinoma: NPC, Laryngeal squamous cell carcinoma: LSCC, thyroid carcinoma: THCA. Ns represents P > 0.05. ∗P < 0.05; ∗∗ P < 0.01; ∗∗∗P < 0.001; ∗∗∗∗P < 0.0001.

Prediction of responses of risk groups to immunotherapy/ chemotherapy

A total of 46 differently expressed immune checkpoint genes were identified. As shown in Fig. 10A, the low-risk group exhibited the highest expression of immune checkpoint genes. Risk score was also negatively correlated with PDCD1 (PD-1), CTLA4, and CD274 (PD-L1; Fig. 10B). In the TCGA-HNSC samples, risk score was positively correlated with TIDE: those with a higher risk score also had a higher TIDE score (Fig. 10C). Additionally, 41 medications were strongly associated with risk score (Fig. 10D). The high-risk group showed higher IC50 values of Cisplatin and 5-Fluorouracil, suggesting that high-risk HNSC patients were more resistant to these medications (Fig. 10E).

Figure 10 Prediction of responses of risk groups to immunotherapy/chemotherapy in TCGA-HNSC dataset.

(A) The expression patterns of 46 differential immune checkpoints between risk groups. (B) Correlation analysis for risk score and representative immune checkpoints (CTLA4, CD274 and PDCD1) as well as the distribution of immune checkpoints between risk groups. (C) Correlation analysis for risk score and TIDE and the distribution of TIDE between risk groups. (D) Correlation analysis for 41 small molecular drugs and risk score. (E) Boxplots of IC50 values of drugs between risk groups, and correlation between the IC50 values and risk score in. Ns represents P > 0.05. ∗∗∗∗P < 0.0001.

Expression of CTLA4, CAMK2N1, PLAU, and CALML5 was enhanced in the OSCC cell lines

The mRNA expressions of CTLA4, CAMK2N1, PLAU, and CALML5 were examined in CAL-27, SCC-25, and HOEC cell lines by qPCR (Figs. 11A–11D). The results showed the expression levels of CTLA4, CAMK2N1, PLAU, and CALML5 were increased in the CAL-27 and SCC-25 cell lines compared to the HOEC cell lines. The risk scores of the HOEC cell lines were significantly higher than the other cell lines (Fig. 11E). Western blot results demonstrated that CALML5 was highly-expressed in CAL-27 and SCC-25 (Figs. 11F–11G). After suppressing the expression of CALML5 in CAL27 and SCC-25 and overexpressing CALML5 in HOEC cells (Fig. 11H), cell viability of CAL27 and SCC-25 cells was detected by CCK8. A significant decrease was observed in the viability of CAL27 and SCC-25 cell lines after inhibiting CALML5, whereas the cellular activity of HOEC was increased after CALML5 overexpression (Figs. 11I–11K).

Figure 11 CTLA4, CAMK2N1, PLAU and CALML5 expressions were enhanced in the OSCC cell lines.

(A) CTLA4 mRNA expression was higher expression in CAL-27 and SCC-25 cell lines. (B) CAMK2N1 mRNA expression was higher expression in CAL-27 and SCC-25 cell lines. (C) PLAU mRNA expression was higher expression in CAL-27 and SCC-25 cell lines. (D) CALML5 mRNA expression was higher expression in CAL-27 and SCC-25 cell lines. (E) The risk scores calculated by the four genes expressions were increased in CAL-27 and SCC-25 cell lines. (F–G) CALML5 protein expression was enhanced in CAL-27 and SCC-25 cell lines. (H) The transfection efficiency of siRNA as well as overexpression plasmids was verified by RT-qPCR. (I) Cell viability of HOEC after overexpression of CALML5. (JI) CAL-27 cell viability was inhibited by si CALML5. K: SCC-25 cell viability was inhibited by si CALML5. ∗P < 0.05; ∗∗P < 0.01; ∗∗∗P < 0.001; ∗∗∗∗P < 0.0001. Mean value of central tendency was used.

Discussion

HNSC affects several organs and is a one of the leading causes of cancer-related death. There is an urgent need to develop a more suitable clinical classification and prognostic signature for predicting HNSC prognosis and guiding precision treatments for HNSC patients. Based on PCD-related genes, this study developed a PCD-related prognostic signature and four molecular subtypes capable of predicting HNSC prognosis and immunotherapy or chemotherapy response.

CTLA4 and PD-1/PD-L1 negatively regulate the immune function of T cells in different stages of T-cell activation (Zhang et al., 2021). It has been reported that the activation of the PD-1/PD-L1 pathway contributes to immunosuppression during anti-tumor therapy in EGFR-driven lung tumors by suppressing T-cell function and increasing the release of pro-inflammatory cytokines (Akbay et al., 2013). An increased proportion of LAG3+ lymphoma cells PD-L2+, PD-1+, PD-L1+ is the predominate cause of immune escape of diffuse large B cell lymphomas (Laurent et al., 2015). PD-L1 is an important immune checkpoint expressed by HNSC, and elevated PD-L1 expression is based on DNA damage response and JAK-dependent pathways (Lailler et al., 2021). This study found the expressions of PDCD1 (PD-1), CTLA4, and CD274 (PD-L1) were significantly elevated in HNSC subtype C4 compared with other HNSC subtypes, which indicates that patients with subtype C4 may be more likely to experience escape from immunotherapy. Some immune-related pathways and inflammatory pathways were also observed in subtype C4, indicating that the increased expression of PD-1, CTLA4, and PD-L1 might induce immune dysfunction and promote an inflammatory response in subtype C4.

HNSC tumor tissues show elevated CTLA4 expression, which is related to OS (Zhang et al., 2020). A possible indicator for CTLA4 treatment is the association between immune cells and CTLA4 expression (Zhang et al., 2020). The effectiveness of immunotherapy and the selectivity of Treg depletion in TME could be improved by maintaining the CTLA4 checkpoint (Liu & Zheng, 2020). The upregulated expression of the epithelial to mesenchymal transition genes CAMK2N1 and WNT5A is associated with the progression of prostate cancer (Peng et al., 2021; Wang et al., 2014; Zhang et al., 2022b). CAMK2N1 is an endogenous inhibitor of calcium/calmodulin-dependent kinase II and has been reported as a tumor suppressor in hepatocellular carcinoma, prostate cancer, and colorectal carcinoma (Carneiro et al., 2019). One previous study demonstrated that the upregulated DEG CAMK2N1 is linked with poor OS and progression-free interval (PFI) of high-risk squamous cell carcinoma patients (Feng et al., 2021). CAMK2N1 is a risk biomarker for HNSC prognosis, while CALML5 is as a favorable prognostic gene for HNSC patients (Chi et al., 2021). Plasminogen can be converted into plasmin through PLAU encoding a secreted serine protease. Highly-expressed coagulation fibrinolysis genes, including PLAU, PLAUR, and SERPINE1, are associated with monocytic infiltration and overexpressed immune checkpoints (PD-L2 and CD276/B7-H3) in pan-cancer, indicating that increased PLAU may be related to the TME in cancer (Saidak et al., 2021). Moreover, Zhang et al. (2018) identified PLAU as one of the hub genes for oral squamous cell carcinoma development and malignant phenotypes. This study first identified four PCD-related genes that were implicated in HNSC: CTLA4, CAMK2N1, PLAU, and CALML. This study also found that these four genes had higher expression levels in HNSC cell lines, and the deletion of CALML5 inhibited OSCC cell viability. These findings suggest that these genes may be involved in the development and progression of HNSC.

This study also found that high-risk HNSC patients were more likely to develop drug resistance to cisplatin and fluorouracil, whereas low-risk patients were more responsive to these drugs. The most successful treatment for HNSC is the combination of docetaxel, cisplatin, and 5-fluorouracil (Goel et al., 2022). An earlier investigation found that CAMK2N1 has low expression levels in Cisplatin-induced resistant SKOV3 cells, suggesting that CAMK2N1 may be involved in Cisplatin-related resistance (Häfner et al., 2016). After undergoing Cisplatin-based neoadjuvant chemotherapy, non-responders have been reported to have elevated expression of the calcium-sensing proteins CALML3, CALML5, and S100A7A (Hepburn et al., 2021). A recent study found that the application of 5-Fluorouracil in HNSC promotes the expression of PD-L1 through the specific DNA damage response and JAK-related pathways, providing a potential therapeutic possibility (Lailler et al., 2021; Zhang et al., 2022a). The PCD-related prognostic signature outlined in this study may help the selection of optimal drugs for HNSC patients.

This study also has some limitations. First, the training set from the TCGA includes primarily white patients, and is not a racially-diverse sample. Although the prognostic value of the PCD-related gene signature introduced in this study was validated in external datasets, it should be further validated through long-term clinical application and prospective studies with larger, more diverse samples. Due to the difficulty of developing clinical immunosuppressive therapy trials, this study also did not include complete cohort data. Four PCD-related genes, CTLA4, CAMK2N1, PLAU, and CALML5 were discovered to be associated with HNSC, but their functional roles in HNSC development and progression remain unknown. More experimental studies are needed to reveal the underlying mechanisms of these genes in HNSC.

Conclusion

Four HNSC-specific PCD-related subtypes were identified, which may help improve clinical stratification for HNSC patients. An HNSC prognostic signature of four PCD-related genes (CTLA4, CAMK2N1, PLAU, and CALML5) was then generated and validated. This prognostic signature could predict HNSC prognosis, TME, and therapeutic responses of HNSC patients. These findings could improve the clinical stratification of HNSC and help guide precision treatments to improve the outcomes of HNSC patients.

Supplemental Information

Supplemental Information 1 Working flow chart

Click here for additional data file.

Supplemental Information 2 A total of 26 PCD-related genes significantly affecting prognosis in TCGA-HNSC patients

Click here for additional data file.

Supplemental Information 3 (A–B), Consensus CDF curves and CDF Delta area in TCGA-HNSC. (C) Clustering heatmap of TCGA-HNSC samples when consensus k =4

Click here for additional data file.

Supplemental Information 4 (A), PPI analysis of 300 DEGs. (B), functional enrichment of 300 DEGs using DAVID software

Click here for additional data file.

Abbreviations

HNSC head and neck squamous

PCD programmed cell death

TME tumor microenvironment

TCGA The Cancer Genome Atlas

GEO Gene-Expression Omnibus

TNM tumor-node metastasis

DEGs differentially expressed genes

LASSO least absolute shrinkage and selection operator

KEGG Kyoto Encyclopedia of Genes and Genomes

GSEA gene set enrichment analysis

CDF cumulative distribution function

PFS progression-free survival

OS overall survival

PCA principal components analysis

ssGSEA single-sample gene set enrichment analysis

ROC receiver operating characteristic analysis

AUC area under ROC curve

stepAIC stepwise Akaike information criterion

FDR false discovery rate

TIDE tumor immune dysfunction exclusion

ICI immune checkpoint inhibition

IC50 half-maximal inhibitory concentration

Additional Information and Declarations

Competing Interests

Author Contributions

Data Availability

The authors declare there are no competing interests.

Chengbo Xu conceived and designed the experiments, analyzed the data, prepared figures and/or tables, authored or reviewed drafts of the article, and approved the final draft.

Hongfang Xu conceived and designed the experiments, performed the experiments, authored or reviewed drafts of the article, and approved the final draft.

Baimei Liu conceived and designed the experiments, analyzed the data, authored or reviewed drafts of the article, and approved the final draft.

The following information was supplied regarding data availability:

The code is available at GitHub and Zenodo:

–https://github.com/BaimeiLiu/code.git.

- BaimeiLiu. (2023). BaimeiLiu/code: v1.0.0 (v1.0.0). Zenodo. https://doi.org/10.5281/zenodo.8415684

The datasets are available at GEO: GSE65858 and GSE41613.

https://www.ncbi.nlm.nih.gov/geo/query/acc.cgi?acc=GSE65858

https://www.ncbi.nlm.nih.gov/geo/query/acc.cgi?acc=GSE41613

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
