# Peer review of "Head and neck squamous cell carcinoma-specific prognostic signature and drug sensitive subtypes based on programmed cell death-related genes"

_PeerJ, doi:10.7717/peerj.16364_

## Round 0.1 · original submission · Minor Revisions

Please see the reviewers' comments, and carefully revise.

·

Basic reporting

Programmed cell death (PCD) as a cell death that includes apoptosis, cuproptosis, etc., it play vital role in the considerable detrimental diseases. Based on PCD-related genes, authors not only divided patients with head and neck squamous cell carcinoma (HNSC) of which subtypes have distinct prognosis and immune score, they also detected 4 PCD-related prognostic signatures to construct prognostic model which has shown well performance in prognosis prediction, the prediction has been validated in external datasets. What's more, it gave the potential therapeutic drugs, and 4 PCD-related prognostic signatures have also validated by cell culture and transient transfection. All in all, it's a comprehensive work with multiple biological analysis. References are the most sufficient, with clear logic and sufficient repeatability

Experimental design

no comment

Validity of the findings

1. The function of the specific signature should be highlight, such as signature for diagnostic or prognostic, etc..
2. In this research, ACLML5 was focus at line 328, but PLAU was discarded. More results about other genes in figure 9H should be expanded.

Additional comments

1. The source of 12 PCD genes at line 111 is unclear, and what's the relation between PCD gene signature and PCD-related gene?
2. In figure 1B, the box of Tumor vs. Normal not in right place, and if the gene locates in this box is not the significant differentially expressed gene, it's better use the dashed line instead of the box.
3. In line 299-301, the 4 genes are filtered from LASSO and stepwise multivariate regression analysis? Make sure the stepwise multivariate regression analysis is one of the methods here.
4. The superlative form at line 332, it's appropriate to use comparative form when comparison between two groups.
5. The incomplete full name of HNSC at line 32. WB at line 350 is an abbreviation? Please provide the full name of the abbreviation.

·

Basic reporting

This study. Overall, it meets the publication standards, but there are still some issues that need to be addressed before publication.
1. The description of the connection between hNSC and PCD is not very clear in the introduction. Please elaborate on why the two should be linked together or introduce the preliminary work.
2. The experimental method should provide a more detailed description of the workflow or add references to help readers better understand, such as line111-116
3. Simplify the language for RT-qPCR and WB experiments, and there is no need to specifically specify non special parameter experiments, such as line189-195
4. Pay attention to language errors. What does line218 A P<0.05 mean?
5. Please present the shortcomings of this experiment in the discussion, rather than simply summarizing the experimental results. There is a lack of discussion on the positive and negative correlation between the expression of CTLA4 and CALML5, and there is no better explanation of the results of the basic experiment. Please supplement.
6. There have been multiple instances of low expression and down call errors in the text, such as CTLA4 being low expression rather than downregulation in FIG11, and such errors have occurred multiple times throughout the entire text.
7. Check the statistical images of FIG11 F and G to ensure the accuracy of the statistical data.

Experimental design

Squamous cell carcinoma of the head and neck is a complex malignant tumor, and research on its occurrence, development, and specific mechanisms has always been a direction of exploration for researchers. The author analyzed the data from the TCGA database and identified differentially expressed genes, thereby identifying TCGA prognostic markers and PCD related prognostic genes. Subsequently, basic experiments such as Western blotting were used to validate these targets. Four prognostic genes were identified to construct a prognostic model. The experimental design is rigorous, the statistical methods are appropriate, and a new understanding of the classification of hNSC has been gained, providing reliable PCD related prognostic indicators for the prognosis prediction and treatment guidance of hNSC.

Validity of the findings

no comment

Additional comments

1. It is recommended to increase the construction of PPI networks and analysis of related pathways in STRING and DAVID databases.

Reviewer 3 ·

Basic reporting

I have reviewed your manuscript and would like to address several points that require attention and improvement.

1. On line 45, "highest immune sore" should be corrected to "highest immune score" to ensure accuracy and clarity in the manuscript.

2. On line 320, it would be more appropriate to state "T cells CD4 memory activated" instead of "cells CD4 memory activated" to accurately describe the specific cell population under consideration.

3. I recommend enlarging the font in most figures, particularly in Figure 1 and Figure 6, to enhance readability and ensure that the details are easily discernible.

4. Regarding Figure 9H, it is crucial to provide the full names and explanations for abbreviations used for cell lines such as OSCC, NPC, and LSCC. Including this information in the article will help readers understand the context and properly interpret the results.

5. In Figure 11, it would be valuable to include the detection of protein expression levels of CALML5 in both the si NC and si CALML5 groups using Western blot or other appropriate experimental techniques. This additional analysis will strengthen the findings and provide a more comprehensive understanding of CALML5's role in the experimental setup.

Experimental design

no comment

Validity of the findings

no comment

Additional comments

The article under review has presented a wealth of data obtained through meticulous analysis of public databases. The authors have demonstrated a comprehensive exploration of the data and have conducted a detailed analysis of their findings. In order to facilitate reproducibility and promote transparency, I kindly request that the authors share the code they utilized in a publicly accessible platform such as Github or Gitlab. By making the code openly available, other researchers will have the opportunity to replicate and validate the results, fostering collaboration and advancing scientific knowledge.

·

Basic reporting

The quality of English language is satisfactory but there are some abbreviations not explained in the main text (SNV, ROC, AUC..) or wrong words and phrases, like "score" instead of "sore" (line 45), "5-year survival rate" instead of "5-year survival" (line 69) or "Wilcoxon test" instead of "Wilcox.tests" (line 215). Also, HNSC should be explained as "head and neck squamous cell carcinoma", not just "head and neck squamous" (line 32).
Literature references are appropriate but there are some errors in reference 15.
Article structure and figures are appropriate, there are no tables while list of primers should be presented in table. Raw data was shared and is available through a private account of the commercial file sharing webpage, what does not guarantee a long-term availability.
Like in almost all publications which are based on re-analyses of TCGA/GEO/etc. datasets, much more bioinformatic analyses are performed but just smaller chunk of results are usually discussed in 'Discussion'.

Experimental design

Like every so-called "TCGA-paper" also this one used common set of bioinfomatic tools, with some minor in vitro confirmations. Unfortunately, study behind this manuscript is quite irreproducible because important details are missing from almost all methods' descriptions.
Therefore, authors must:
1) Provide version numbers for ALL used R packages.
2) Provide valid URLs and cite references for ALL used web-based tools and databases.
3) State precisely ALL used R packages, especially those used for inferential statistics, survival analyses (KM and Cox), drawing graphs (violin plots, Venn diagrams, PCA...), etc.
4) State the exact model and manufacturer of ALL used instruments (microplate reader, WB imager...).
5) Cite original references for ALL used datasets (TCGA-HNSC, GSE65858 and GSE41613).
6) Explain how reverse transcription was performed and how relative expression was calculated (citing also PMID: 11846609 for 2^-ddCt method).
7) From Cellosaurus database, state Resource Identification Initiative ID (RRID) for all used cell lines and clearly state that both used HNSC cell lines represent tongue squamous cell carcinoma.
8) Provide used dilution, catalog number and manufacturer also for secondary antibodies, as well as what was used for WB visualization (which chemiluminiscent kit).
9) Provide P-value for each AUC, and state which post hoc test was used with Kruskal-Wallis test. Also state which measure of central tendency and dispersion was presented in Figure 11.

Validity of the findings

Like every so-called "TCGA-paper" also this one is primarily descriptive and merely theoretical, and authors brought conclusions based on those premises. In vitro add-ons are welcomed but statement in the 'Abstract' "Cell counting kit-8 (CCK-8) test, Western blot, and realtime reverse transcriptase-polymerase chain reaction (RT-qPCR) were used to evaluate the expression and function of prognostic genes." is simply NOT true because with just measuring cell viability you cannot evaluate the function of a gene, much more in vitro and in vivo experiments are needed for that.
Another completely wrong statement is "head and neck squamous [cell carcinoma] (HNSC) is a main cause of cancer mortality" (lines 32-33) since this type of cancer is only on 17th place by mortality rate (https://seer.cancer.gov/statfacts/html/common.html).
Authors should also keep in mind, and accordingly adapt their text, that markers with AUC 0.6 to 0.7 are not considered very discriminating (i.e., useful) (PMID: 20736804).
To be honest, exactly the same conclusions presented in this manuscript could be brought with much less bioinformatic analyses.

Additional comments

There are some more drawbacks which must be further improved or corrected:
1) Since authors used only TCGA-HNSC dataset, this must be always clearly presented, so proper phrase would be like "TCGA-HNSC prognostic signature".
2) Lines 61-63: State to which country, world, etc. those numbers relate to.
3) Explain how were data for "para-carcinoma tissues" obtained and state number of samples.
4) Line 179: Explain what is "|cor|".
5) Line 186: It is unclear what PPARG siRNA was used for.
6) Line 209: Since you had treatment only with siCALML5, you cannot claim "various treatments". Also, a multiplication sign is missing from the number of seeded cells.
7) Explain in more detail what are "PCD-related genes" since just a reference is not enough.
8) It is unclear what means "co-DEGs".
9) It would be very useful for an easier orientation through this manuscript if authors provide a scheme of their study (used datasets, sample numbers, tools, databases...obtained results).

---

## Round 0.2 · Minor Revisions

I think the authors have response all the Reviewers' Comments. It is a good revision, and indicated a specific prognostic signature for development and treatment of head and neck squamous cell carcinoma.

Before the manuscript can be accepted, it needs editing for English.

For example, the abstract could be edited to:

> Background: As a complex group of malignancies, head and neck squamous cell carcinoma (HNSC) is one of the leading causes of cancer mortality. This study aims to establish reliable clinical classification and a gene signature for HNSC prognostic prediction and precision treatments. Methods: According to genes linked to programmed cell death (PCD), consensus clustering analysis was performed to group HNSC patients in The Cancer Genome Atlas (TCGA) database. We discovered differentially expressed genes (DEGs) between subtypes using the "limma" R package. The TCGA prognostic signature and PCD-related prognostic genes were found using the least absolute shrinkage and selection operator (LASSO) regression analysis and univariate Cox regression analysis. The robustness of the LASSO analysis was validated using datasets GSE65858 and GSE41613. A cell counting kit-8 (CCK-8) test, Western blot, and real-time reverse transcriptase-polymerase chain reaction (RT-qPCR) were used to evaluate the expression and viability of prognostic genes. Results: Four molecular subtypes were identified in PCD-related genes. Subtype C4 had the best prognosis and the highest immune score, while subtype C1 exhibited the most unfavorable outcome. Three hundred DEGs were identified among four subtypes, and four prognostic genes (CTLA4, CAMK2N1, PLAU and CALML5) were determined to construct a TCGA-HNSC prognostic model. High-risk patients manifested poorer prognosis, more inflammatory pathway enrichment, and lower immune cell infiltration. High risk patients were more prone to immune escape and resistant to Cisplatin and 5-Fluorouracil. Prognosis prediction was validated in external datasets. The expression of CTLA4, CAMK2N1, PLAU and CALML5 was enhanced in CAL-27, SCC-25 cell lines, and CALML5 inhibited CAL-27, SCC-25 cell viability. Conclusion: Our study shares novel insights into HNSC classification and provides a reliable PCD-related prognostic signature for prognosis prediction and treatment for patients with HNSC.

This is only an example - the entire manuscript should be edited.

**Language Note:** The Academic Editor has identified that the English language must be improved. PeerJ can provide language editing services - please contact us at copyediting@peerj.com for pricing (be sure to provide your manuscript number and title). Alternatively, you should make your own arrangements to improve the language quality and provide details in your response letter. – PeerJ Staff

·

Basic reporting

None

Experimental design

None

Validity of the findings

None

Additional comments

None

·

Basic reporting

no comment

Experimental design

no comment

Validity of the findings

no comment

Additional comments

no comment

Reviewer 3 ·

Basic reporting

The concerns of reviewer 3 have all been answered. The paper is ready to publish.

Experimental design

The concerns of reviewer 3 have all been answered. The paper is ready to publish.

Validity of the findings

The concerns of reviewer 3 have all been answered. The paper is ready to publish.

Additional comments

The concerns of reviewer 3 have all been answered. Thank you for making the code public. The paper is ready to publish.

·

Basic reporting

Authors have satisfactorily responded to all my concerns and questions, and appropriately improved quality of this manuscript.

Experimental design

Authors have satisfactorily responded to all my concerns and questions, and appropriately improved quality of this manuscript.

Validity of the findings

Authors have satisfactorily responded to all my concerns and questions, and appropriately improved quality of this manuscript.

---

## Round 0.3 · Minor Revisions

The English still requires editing. The Language Editing Certificate must be uploaded in the submission system.

**Language Note:** The Academic Editor has identified that the English language must be improved. PeerJ can provide language editing services - please contact us at copyediting@peerj.com for pricing (be sure to provide your manuscript number and title). Alternatively, you should make your own arrangements to improve the language quality and provide details in your response letter. – PeerJ Staff

---

## Round 0.4 · accepted · Accept

I think that the current version is ready to be accepted for publication.